# Automated Support for Battle Operational–Strategic Decision-Making

**Gerardo Minguela-Castro \***[ID]**, Ruben Heradio**[ID] **and Carlos Cerrada**[ID]

Department of Computer Systems and Software Engineering, Universidad Nacional de Educacion a Distancia (UNED), 28040 Madrid, Spain; rheradio@issi.uned.es (R.H.); ccerrada@issi.uned.es (C.C.)
\* Correspondence: gminguela1@alumno.uned.es or gminguela@isdefe.es

**Abstract:** Battle casualties are the subject of study in military operations research, which applies mathematical models to quantify the probability of victory vs. loss. In particular, different approaches have been proposed to model the course of battles. However, none of them provide adequate decision-making support for high-level command. To overcome this situation, this paper presents an innovative high-level decision-making model, which uses an adaptive and predictive control architecture. The paper reports empirical evidence supporting our model by considering one of the greatest battles of World War II: the Battle of Crete.

**Keywords:** decision support systems; combat models; system dynamics; battle situation; warfare information system

## 1. Introduction

Lanchester's seminal work [1] on battle dynamics' modeling has inspired significant research on the development of combat abstractions to support military decision-making under uncertainty, pursuing how to achieve superiority in combat. Lanchester's original model and its distinct evolving extensions have dominated the dynamic assessment of conventional land force balance for a long time [2], being used by major organizations (e.g., the US Army, the Office of the Secretary of Defense, etc.) to assess a wide variety of issues (e.g., evaluating the balance of operation theater [3,4], guiding decision on weaponry choices [5], etc.).

Nevertheless, it is worth noting that Lanchesterian models have important limitations, e.g., they perform an oversimplistic one-side treatment without taking into account the opponent's capabilities, and they cannot be used for disaggregated engagements.

Another matter to be taken into account is the abstraction level supported by the decision-making procedures. Military doctrine usually distinguishes the following three levels of command:

1. The strategic level studies the conflict from the most abstract perspective, considering the war final outcomes as a whole. It involves the overall planning, resource distribution, and organization of the military force. Additionally, it defines and supports the national policy.
2. War is divided into campaigns, which are organized into operations. The operational level deals with the design, arrangement, and execution of campaigns and principal operations.
3. The Tactical level implements the campaign operations on the battlefield.

Interestingly, most decision-making approaches, including the non-Lanchesterian ones, are focused on the tactical level of command [6,7]. In other words, the operational and strategic levels of command are insufficiently supported by existing decision-making systems.

This paper proposes an innovative framework that overcomes most limitations of Lanchesterian models and supports decision-making at the highest command levels: the

strategic and the operational ones. Our framework applies adaptive and predictive control engineering methods to dynamically adjust to changes in the battle, taking into account the capabilities and maneuvers of the adversary and the effects produced. Additionally, it includes a learning mechanism to improve decisions under conditions with high uncertainty.

Finally, the paper reports the empirical evaluation of our framework on the Battle of Crete, an influential World War II battle where paratroopers were used massively for the first time. This, by itself, constitutes a relevant contribution as most literature on military decision-making lacks adequate experimental validations. In particular, most validations follow mathematical procedures that make non-realistic assumptions [8] or rely on simplistic made-up examples [6,9].

The remainder of this paper is organized as followings. Section 2 escribes our framework and Section 3 reports its empirical validation. Finally, Section 4 provides some concluding remarks and discusses future challenges.

## 2. A Framework to Support Battle Operation-Strategic Decision-Making

There are two principal battle analysis mechanisms alternative to classical Lanchester's models: (i) stochastic models and (ii) deterministic models, some of them in the Lachesterian tradition [10,11]. Currently, other approaches such as intelligent agents are gaining substantial momentum [12,13]. These new models aim to extend the capabilities [6,9] and reduce the shortcomings of previous approaches [14,15]. However, they fail to be an appropriate benchmark for high-level decision-making.

Our framework overcomes the limitations of Lanchester's original work, which are profoundly discussed in [16], by treating the battle as a cause-effect process that evolves according to the dynamics of the Lanchester's equations subject to changes and external actions. To do so, our approach applies the adaptive and predictive control theory introduced in [17], which incorporates uncertainty modeling techniques. Our approach architecture comprises a set of blocks that work cooperatively and ensure that decision-making is carried out coherently, following the military doctrine. In particular, a set of sequential stages trigger the definition of the applicable strategy, the evaluation and selection of the different possible COAs, and the adaptation of the model to the evolution of the operation. Each block represents the mechanics of military thinking, see Figure 1, where $x(t)$ and $y(t)$ define the number of combatants of the x-force and y-force at each instant, $x(t+1)_e$ and $y(t+1)_e$ are the estimated the number of combatants for the following instant.

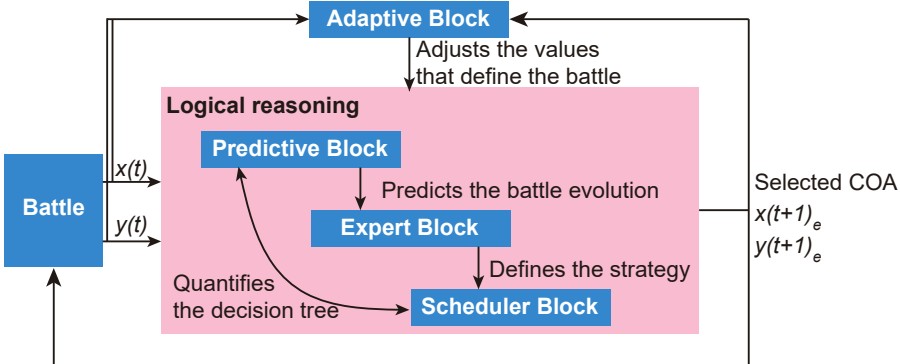

**Figure 1.** Architectural design of our framework. Each block represents the mechanics of military thinking, thus (i) assessing the events of the battle that will define the strategy to be followed and selecting the COA to accomplish the mission, (ii) identifying the resources that will be necessary to carry it out, and finally (iii) adapting to the outcomes.

The implementation requires a logical process capability and should simulate the decision-making process, from prediction to action. In this context, the new framework is

formulated and tested (it will be robust if its application on real confrontations meets the expectations in terms of performance and consistency).

Figure 2 develops the essential elements that trigger the choice of a specific COA. The predictive block generates the control signal (prediction). The adaptive block adjusts the parameters of the constituent blocks based on the deviation of the output signal (the actual situation) from the predicted one. The expert block acts trying to modify the trend defined by the predictive block through the scheduler block, thus changing the course of actions following the needs of the battle. It is worth noting that the action development times are operation times and that the available databases with information on conflicts are usually represented by time evolutions in days, in the best case.

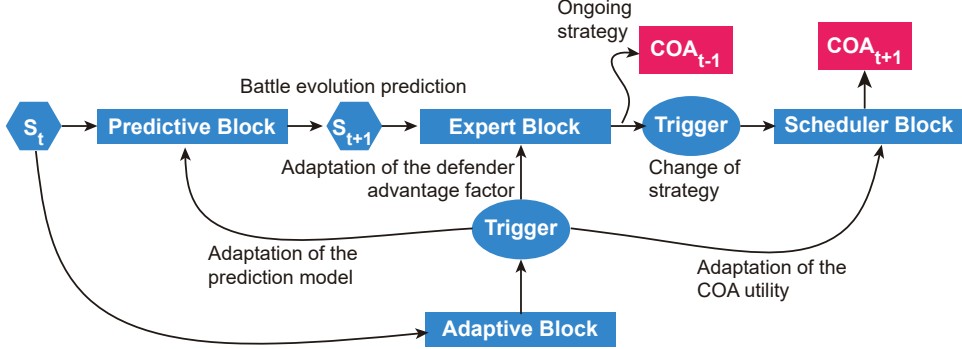

**Figure 2.** Primary elements that trigger the choice of a specific COA in the new framework through a sequential model.

### 2.1. Predictive Block

In military doctrine, intelligence is defined as the interpretation and integration of knowledge about the terrain, meteorology, population, activities, capabilities, and intentions of a present or potential enemy. The intelligence cycle is composed of the phases of direction, acquisition, elaboration, and dissemination. The predictive model will recreate this cycle in the prediction of scenarios necessary to evaluate future decision-making. Tactics, combat strength, and attrition are identified as the most critical factors for modeling the dynamic prediction of a future confrontation. The predictive block defines the future trend of the confrontation at an instant after the current one using the Lanchester equations and a regression model.

#### 2.1.1. Lanchester Models

Lanchester models are defined as Ordinary Differential Equations (ODEs) that support the prediction of confrontation results. These equations simplify the battle models, pointing the importance of troop concentrations up in the final result. They consider two confronted forces, denoted as $x$ and $y$. For simplicity, forces are typically modeled as the number of combatants, i.e., the size of each army. Thus, $x(t)$ and $y(t)$ define the number of combatants of the $x$ and $y$ forces at instant $t$. Additionally, Lanchester's equations usually consider each force's lethality, denoted as $a$ and $b$, whose calculation depends on the fire and combat typologies.

Table 1 summarizes the most prominent Lanchester equations for combat, assuming no reinforcements, according to their fire type and the degree command and control maintained by the command of the situation. The following types of fire are considered:

- *Direct (aimed) Fire*: Each member of the x-force is within the range of the enemy and, when the x-force receives casualties, the fire is concentrated on the remaining ones. See Lanchester [1].
- *Fire Concentrated in areas*: In the case of forces distributed in areas invisible to the enemy or using concentrated fires in areas such as artillery, the model of casualties of the x-force must be proportional to the size of the x-force. See Lanchester [1].

- *Combat Asymmetric*: battles between conventional x-force forces against guerrilla y-forces (invisible to the enemy). See Deitchman [18].
- *Unequal sized forces*: The difference in size between the contenders is a factor that conditions the lethality and inefficiencies of scale. Therefore, Helmbold [19] added $E_x$ and $E_y$ functions that modify force lethality by a $x$ and $y$ ratio.
- *Great Battle*: A campaign on a big scale, i.e., an aggregation of many smaller battles. See Fricker [20].

**Table 1.** Summary of Lanchester equation for combat.

| | | Fire Type | | | | |
|---|---|---|---|---|---|---|
| | | **Direct Fire** *(Square Law)* | **Fire Concentrated in Areas** | **Asymmetric Combat** *(Linear Law)* | **Unequal-Size Forces** | **Great Battles** *(Logarithmic Law)* |
| **Control & Command Level** | **Efficient** | $\frac{dx}{dt} = -ay(t)$ $\frac{dy}{dt} = -bx(t)$ | | | $\frac{dx}{dt} = -ay(t)E_y\left(\frac{x}{y}\right)$ $\frac{dy}{dt} = -bx(t)E_x\left(\frac{y}{x}\right)$ | |
| | **Not so efficient** | | $\frac{dx}{dt} = -ay(t)x(t)$ $\frac{dy}{dt} = -bx(t)y(t)$ | $\frac{dx}{dt} = -ay(t)$ $\frac{dy}{dt} = -bx(t)y(t)$ | | |
| | **Poor** | | | | | $\frac{dx}{dt} = -ax(t)$ $\frac{dy}{dt} = -by(t)$ |

There is no reason all types of fires should not be used together or developed in different phases in a battle. Applying the generalized model defined by Bracken [21] into our approach (Equations (1) and (2)), it is possible to determine the nature of the battle empirically. We define $p$ as the exponential factor of the attack force and $q$ as the exponential factor of the defense force. Where *f(t)* and *g(t)* are the replacement forces or evacuated forces according to the sign.

$$\frac{dx}{dt} = -ay(t)^p x(t)^q + f(t) \tag{1}$$

$$\frac{dy}{dt} = -bx(t)^p y(t)^q + g(t) \tag{2}$$

Defining $p$ and $q$ in the interval [0, 1],

- If $p$ and $q$ is (1, 1) defines the linear law.
- If $p$ and $q$ is (1, 0) it defines the quadratic law.
- If $p$ and $q$ is (0, 1) it defines the logarithmic law.

It is work remarking that the tactical parameter $d$ (offensive or defensive strategy) of Bracken's model [21] is not taken into account because it does not contribute substantially to the adjustment of parameters.

2.1.2. Generalized Regression Model

Regression attempts to explain the causality of the effects. The generalized model [21] generates four variables to be solved. Using (i) the least-squares method as target function and optimized by the Generalized Reduced Gradient (GRG) algorithm, whose mathematical structure is presented by Abadie [22], from data obtained during the course of the battle, and (ii) the following metrics that account for the regression model quality: Sum of Squares Regression (SSR), Sum of Squares Total (SST) and $R^2$, it obtains a feasible estimation procedure to solve the four unknown variables. Therefore, the GRC algorithm manages the slope of the target function as the input values change and determines that it has reached an optimal solution when the partial derivatives are equal to zero. A higher $R^2$ value indicates a better fit for the mean daily losses (estimated attrition). A perfect fit would be an $R^2$ of one.

### 2.2. Expert Block

The development of decision-making is characterized using intelligence resources through the predictive block and its interpretation, leading to the strategy definition. Once the global situation informed by the predictive block has been evaluated, it is necessary to redefine the strategy when there is a change of trend or when such trend change is sought by modifying the strategy (Defensive, Offensive, Stability, etc.).

The large units, in their advanced movement, make contact progressively. The awareness of the adversary's intentions, in specific areas, allows the selection of an adequate strategy, given the general attitude of the adversary:

- A Defensive Battle means a high risk of being attacked and inferiority of resources.
- An Offensive Battle means a low probability of being rejected and superiority of resources.

If the previous operational decisions are within the acceptable limits of attrition, the re-evaluation will not make sense in the first approximation.

Intention Model

The assessment of the adversary's intentions will be based on the actual ability to reject a possible attack in a hostile scenario and the awareness of the enemy's intentions. The contenders will consider a stable state situation if the probability of a failed attack exceeds the security level. Figure 3 depicts a decision tree for evaluating the adversary's intentions.

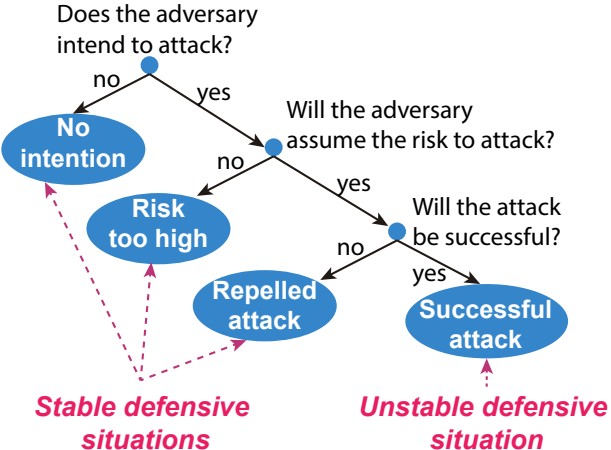

**Figure 3.** Decision tree on adversary intentions in a bipolar situation for assessment, shown in Christensen's report [23]. If the relationship between one's own forces and the adversary is friendly, the adversary will not consider military aggression. On the other hand, if the relationship is hostile, the adversary may wish to attack, and one's own forces may need a military defense against the adversary.

The intention of an opponent to attack will be given by the minimum probability of success that the opponent needs to launch an attack *P* (this figure depends on the doctrine of the contender) and by the probability of being rejected by the defender *WinsDef*, conditions (3) and (4). Probabilities are defined unilaterally through the opponent's vision, so if the adversary requires a high chance of success of the attack before launching it, the *WinsDef* should be low.

- Equation (3) identifies a high risk of being attacked:

$$P < (1 - WinsDef) \tag{3}$$

- Equation (4) identifies a high risk of being rejected:

$$P > (1 - WinsDef) \tag{4}$$

The *WinsDef* curve represents the probability of being rejected by the defender (Equations (5)–(8)), and it is obtained using logistic regression. This allows estimating the probability of success or failure as a function of the defender's Advantage Factor $v$ as defined in Helmbold's work [24], from a subset of data obtained from the CDB90 data set of individual battles, from 1600-1979 (https://github.com/jrnold/CDB90, last visited 26 June 2021). See Figure 4.

Lanchester's Square Law defines the Factor $v$, where $x(0)$ and $y(0)$ are the number of combatants of the x-force attacker and y-force defender at the initial instant, $a$ is the lethality of the defender force, and equivalently $b$ of the attacker. Accordingly,

$$\text{WinsDef} = \frac{1}{1 + e^{0.12 - 3.38v}} \tag{5}$$

$$v = \ln\sqrt{\frac{\delta}{\alpha}} \tag{6}$$

$$\alpha = b\left(\frac{x_0}{y_0}\right) \tag{7}$$

$$\delta = a\left(\frac{y_0}{x_0}\right) \tag{8}$$

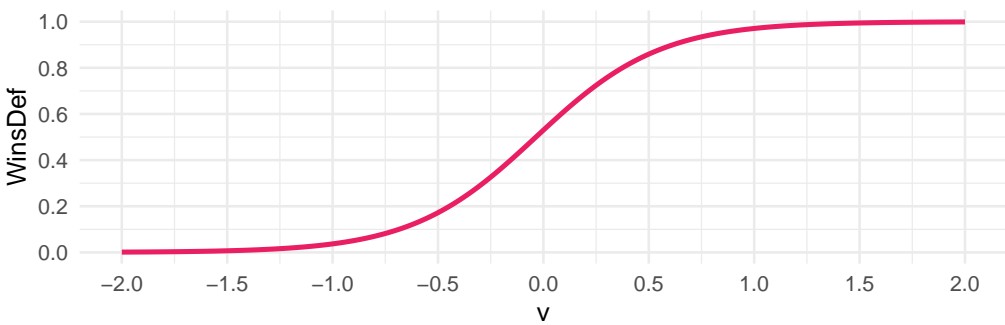

**Figure 4.** Relationship between $v$ and the probability of being rejected by the defender. Empirical evidence shows that the advantage factor favoring the defender has an important influence in determining which side wins, according to Helmbold's work [25].

### 2.3. Scheduler Block

The COA planning is determined by military doctrine and the different factors of the operational environment, such as, for example, the enemy centers of gravity (COGs). Within the military decision-making process, the planning phase involves COA analysis, comparison, and evaluation, as well as the development of the matrix plan that provides the resources and conditions to optimize and maximize the results.

Action planning is inferred through decision trees, which process the doctrinal knowledge (friend and enemy), the strategy defined from the expert block, and evaluate possible outcomes in the context of probable enemy actions obtained through the predictive block.

#### 2.3.1. Alternative Assessment

The assessment of the alternatives is based on the concept of expected value $E(x)$, applicable to random variables that take numerical values and the utility of the COA. The final objective of the selected COA will be the fulfillment of the mission. In the current battle decisions, the own casualties $x$ in combat is the main conditioning factor, so the Wald or pessimistic criterion is taken: it is a question of assuring conservative casualties (MAX MIN), Equation (9). This criterion involves selecting an alternative whose expected or average attrition is lower.

$$\text{COA}_i = min(E(x)) \tag{9}$$

### 2.3.2. Centers of Gravity

All aspects of planning depend on the determination of well-defined, achievable, and measurable objectives. The process of identifying and defining objectives involves knowing the enemy, geography, and climate of the area of responsibility.

The objective acquisition model will be simplified using the K-Means clustering method (by the tactical disposition of the units in the terrain, using the Euclidean distance as a quantitative variable), obtaining the centers of concentration of the deployed units.

K-Means works by finding clusters with a spherical or convex shape, and needs as input data the number of groups in which we are going to segment the population into k cluster, Elbow method, algorithm according to Bholowalia et al. [26], iterates with different values from 1 to n in the sense of reduction of the total sum of intracluster variance. Therefore, for each iteration, it takes the Euclidean distance between each unit with its center and adds up all the squares of the differences calculated (SSE), up to find the elbow point, where the SSE vs. cluster curve rate of decline is sharpened. Figure 5 shows a practical example of the application of the K-Means plus elbow method algorithm for the determination of Japan Centers of Gravity (COGs) in the battle of Manchuria on 8 August 1945.

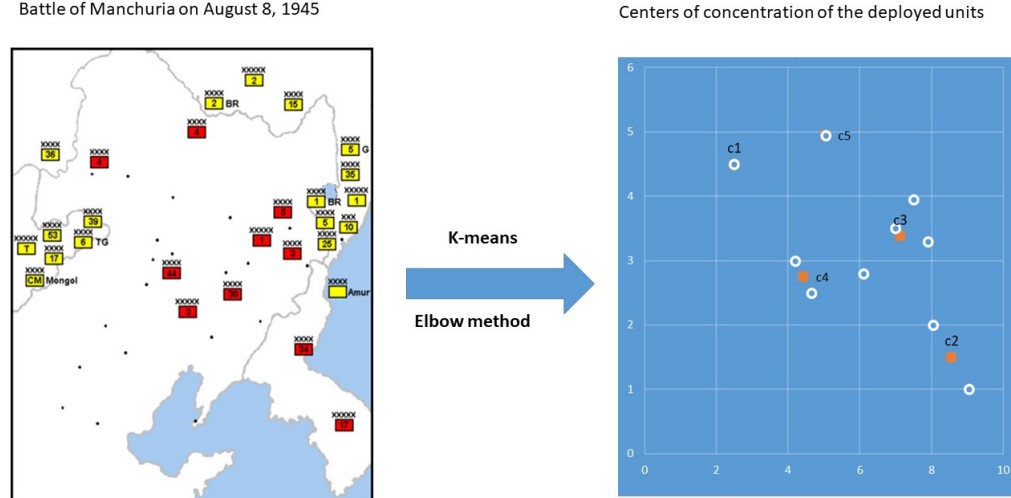

**Figure 5.** The figure on the left depicts the situation described in the biography of the battle of Glantz [27] between Japanese (red) and Russian (yellow) forces. The right-hand figure shows the COGs obtained by applying K-Means plus Elbow method.

### 2.4. Adaptive Block

Even if a good battle model is available, changes in combat dynamics will lead to the deterioration of the model's fit (prediction and driving). Our framework adapts to varying circumstances in the theater of operations and generates changes in the parameters that reflect the decisions' prediction and conditioning. Thus, adaptive control provides a solution theoretically capable of approximating the dynamics of the battle.

The adapting mechanism involves the following tasks:

- Adapting the prediction and factors that determine the strategy to the current battle situation.
- Setting the parameters of the COA usefulness.

This adapting mechanism is a learning process and will provide information for improving the model fit.

Adapting Mechanism

The design of the adapting mechanism has focused on optimizing model prospect (i.e., on error minimization) and improving computational performance.

As Figure 6 shows, an autotune control is used for the predictive block, whose time window is updated step by step with the latest samples. This makes it possible to adapt the values that define the battle to the different phases of the battle, eliminating the jumps produced by random errors or outliers, and the poor information for parameter adjustment in the measurements around an initial operating point.

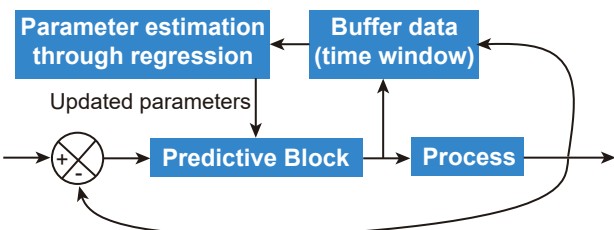

**Figure 6.** The adaptive autotune control recursively estimates the parameter values of the predictive model. The most important aspect of this type of control is having a sufficiently robust parameter estimation technique.

A supervised learning mechanism is used for the expert block adaptation, which extends the binary values (final result) of previous battles and calculates the logistic regression base of the intention model according to the advantage factor. Adaptation is carried out after the final outcome.

Finally, as Figure 7 shows, in the case of the scheduler block, a utility function is used as an adaptation measure to represent the effectiveness in taking planned actions (Friendly Options) by casualty ratio. Feeding the effectiveness of the previously selected COAs concerning the opponent's actions, it will provide the new framework approach with a discarding capability for future tree constructions, avoiding the selection of inefficient COAs.

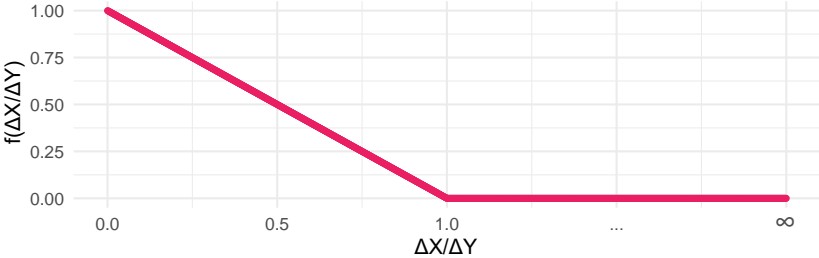

**Figure 7.** *Utility* is a function that relates casualties among opponents $U_i = f(\triangle x / \triangle y)$, where $\triangle x$ stands for own casualties, and $\triangle y$ for enemy casualties. It is worth noting that the utility function is close to 1 for COAs that maximize enemy casualties and minimize their own.

## 3. Empirical Validation

This section reports the empirical validation of our framework on the Battle of Crete, which is one of the greatest battles of the Second World War, where the type of combat was mainly land-based. This mode of combat has not essentially changed since then. Therefore, our experimental results should extrapolate adequately to present-day combat.

In particular, our validation goal is to identify the best possible courses of action and determine the effects they produce on the adversary in comparison with the actual battle on 27 April 1941.

### 3.1. Historical Overview

On 27 April 1941, A. Hitler ordered to invade the island of Crete. Airborne troops carried out the operation under the command of General K. Student, involving 700 transport

planes and 750 gliders supported by the Luftwaffe. The island's invasion was undertaken by 22,000 German paratroopers and mountain troops, and 2700 Italian troops, who took less than two weeks to occupy it. The Allies had 42,547 men of different nationalities. The British evacuated their positions protected by the Royal Navy, which suffered heavy losses. Crete remained in German hands until its garrison capitulated in May 1945. Crete remained in German hands until its garrison capitulated in May 1945. According to the historical data:

- 8100 German paratroopers landed on the first day, 7400 on the second day, and 9500 more evenly over the following days.
- In the different areas of Crete there were deployed: 27,550 British Empire soldiers, 13,000 Greek soldiers, as well as unarmed 4000 to 5000 Cypriots and Palestinians.
- A total of 950 British soldiers landed on the eighth day of the battle.
- Two Greek battalions left the battle when their armament and ammunition ran out, evenly from the third day of the fight. Another 2800 Greek soldiers were captured or killed.
- Approximately 4000 British troops were evacuated on the tenth day of the fighting, and another 11,000 evenly through the thirteenth day of the battle, and 1000 more on the thirteenth day.
- Germans estimated their casualties at 6000, while the British estimated 9000 Germans wounded and 6000 killed.
- There were 2600 British and 2600 Greek soldiers dead. Additionally, 10,500 British and 5600 Greek soldiers were captured.

### 3.2. Battle Analysis

This section describes the dataset, and then summarizes the analysis our framework provides.

### 3.2.1. Dataset

Table 2 and Figure 8 describe the dataset regarding the landing of German troops and the landing or withdrawal of Allied troops during the invasion of Crete. This dataset was obtained from the combined study of the following literature sources: Engel [28], Biank [29], Cox [30] and Miller [31]. Additionally, it is assumed that Lanchester's Square Law is fulfilled for the acquisition of intermediate casualty data not available in the literature. Please note that numbers are divided by 1000. For example, the first row in Table 2 shows that at the beginning of the first day, 40,550 Allied troop soldiers and 8100 German paratroopers landed at Crete in a naval manner. A negative value in the departure column means that new troops landed, e.g., at the beginning of the 7th day, there were a total of 18,187 German and 28,431 Allied soldiers on the island, and 1357 additional paratroopers and 600 new Allied soldiers landed.

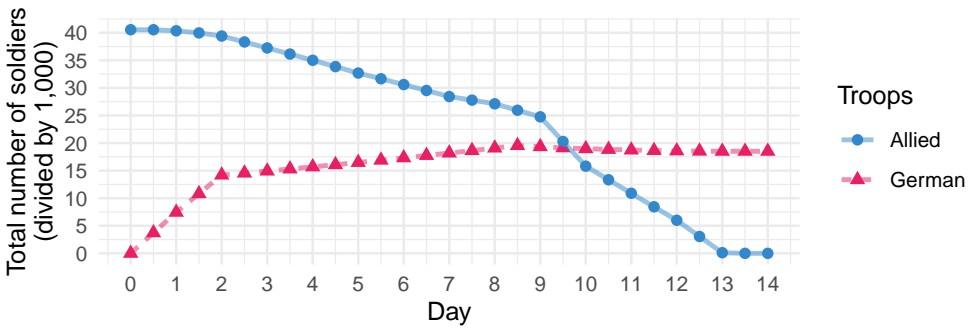

**Figure 8.** German and Allied troops evolution. As the German troops managed to transport enough units to defeat the garrison, Allied troops progressively lost the battle. Although the invasion was successful, there were heavy casualties among the German troops.

**Table 2.** Crete Battle dataset gathered from [28–31]. Data imputation was performed assuming Lanchester's Square Law. Decimal notation is used dividing actual numbers by 1000. *G* and *A* stand for German and Allied.

| | Total Number | | Instant Change | |
| Days | G. Troops | A. Troops | G. Paratroopers | A. Toops Departure |
|---|---|---|---|---|
| 0 | 0 | 40.55 | 8.1 | 0 |
| 0.5 | 3.721545 | 40.55 | 0 | 0 |
| 1 | 7.44309 | 40.35647966 | 7.4 | 0 |
| 1.5 | 10.81620251 | 39.96943898 | 0 | 0 |
| 2 | 14.19245006 | 39.40699645 | 1.357 | 0.667 |
| 2.5 | 14.55175339 | 38.33548905 | 0 | 0 |
| 3 | 14.91973593 | 37.24529787 | 1.357 | 0.666 |
| 3.5 | 15.29654901 | 36.1364716 | 0 | 0 |
| 4 | 15.68234359 | 35.00805105 | 1.357 | 0.667 |
| 4.5 | 16.07727838 | 33.85906919 | 0 | 0 |
| 5 | 16.48151992 | 32.68955071 | 1.357 | 0.35 |
| 5.5 | 16.89523456 | 31.65751167 | 0 | 0 |
| 6 | 17.31730871 | 30.60395948 | 1.357 | 0.35 |
| 6.5 | 17.74791664 | 29.52845942 | 0 | 0 |
| 7 | 18.18723612 | 28.43056776 | 1.357 | −0.6 |
| 7.5 | 18.63544852 | 27.78483148 | 0 | 0 |
| 8 | 19.08889139 | 27.11578816 | 1.357 | 0.35 |
| 8.5 | 19.5477535 | 25.94816581 | 0 | 0 |
| 9 | 19.33757336 | 24.75668262 | 0 | 6.95 |
| 9.5 | 19.13704423 | 20.27612881 | 0 | 0 |
| 10 | 18.97280759 | 15.80600251 | 0 | 2.95 |
| 10.5 | 18.84477897 | 13.34441651 | 0 | 0 |
| 11 | 18.73668919 | 10.88948801 | 0 | 2.95 |
| 11.5 | 18.64848434 | 8.440180169 | 0 | 0 |
| 12 | 18.58011888 | 5.995458984 | 0 | 3.95 |
| 12.5 | 18.53155566 | 3.054292802 | 0 | 0 |
| 13 | 18.50681589 | 0.115651907 | 0 | 0 |
| 13.5 | 18.50587911 | 0 | 0 | 0 |
| 14 | 18.50587911 | 0 | 0 | 0 |

### 3.2.2. Predictive Block

Given the aggregated values in Table 2, the predictive block defines which parameter values (*p*, *q*, *a*, and *b*) fit best the data, using a generalized version of the Lanchester Equations (1) and (2) provided by [21].

In this dynamic process, parameter values are adjusted by the adaptive block, step by step through the battle's evolution, according to the time window selected (7 sample-equivalent to 3 battle days). The procedure used determines the parameter values, applying the generalized regression model depicted in Section 2.1.2.

Figure 9 shows how the values fit around the target values that define the entire battle shown in Table 3 at each iteration, obtaining values of $R^2$ close to 1. This demonstrates that the treatment of major battles must be done in phases due to changes in the troop lethality, as well as the variation in the typology of the confrontations and armaments used, agreeing with other authors as Lucas et al. [32], Rubio-Campillo [33] and Chen [8].

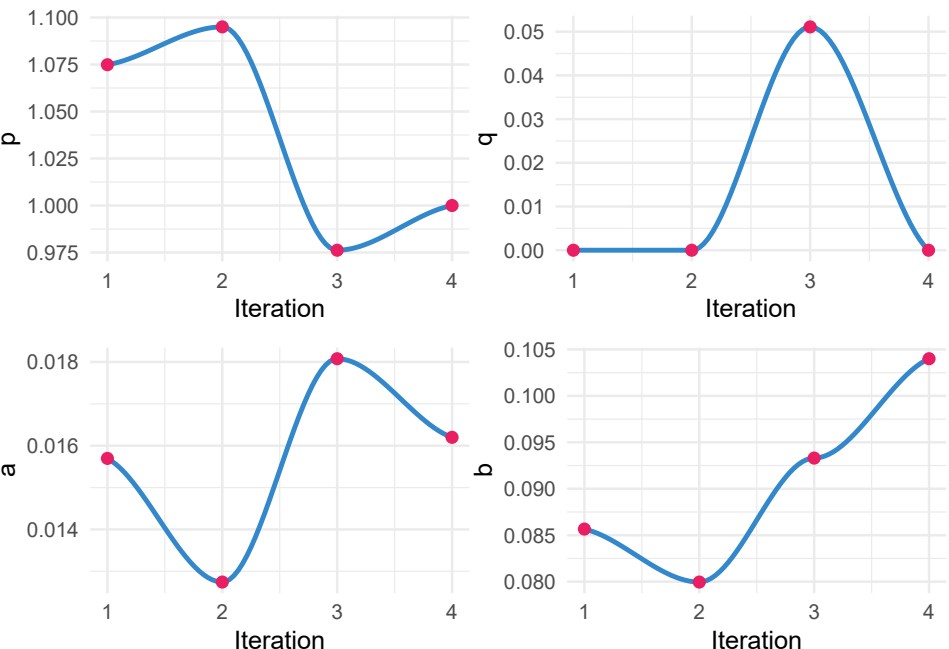

**Figure 9.** Parameter values evolution by adaptive block. Fitted values roughly coincide with the actual ones in the dataset.

**Table 3.** Target parameter values obtained from Engel [28] (Data source: https://apps.dtic.mil/sti/citations/AD0298786, last visited 26 June 2021).

| Parameter | Calculated Values |
|:---:|:---:|
| $p$ | 1 |
| $q$ | 0 |
| $a$ | 0.0162 |
| $b$ | 0.104 |

### 3.2.3. Expert Block

The assessment of German Troop intentions, following the procedure depicted in Section 2.2, identifies a high risk of attack on Allied troops, the probability of success $P$ that German troops need to launch an attack is much lower than that obtained in the assessment. See Figure 10.

$$P_{Germantroops} < (1 - WinsDef) \tag{10}$$

The German Troop intentions allow the selection of an adequate strategy for the Allied troops. In this case, a Defensive strategy is chosen due to the following principles:

- Principle of concentration. The side of the opponent with the greater strength, all other factors being equal, will inflict the greater damage.
- Law of the casualty distribution. The opponent with greater strength will be the one that receives fewer casualties.

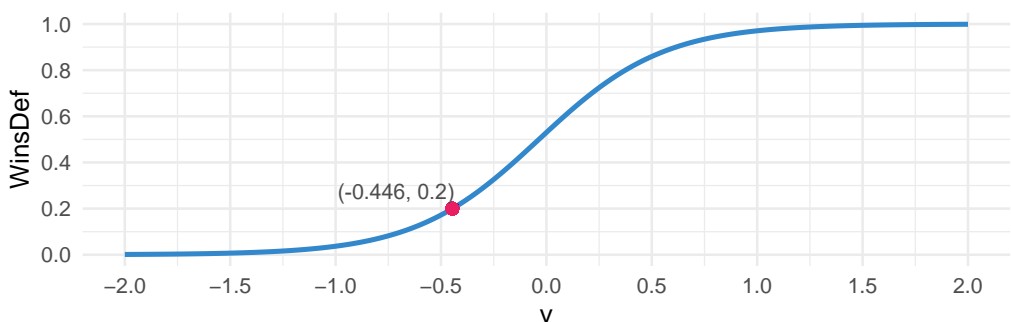

**Figure 10.** Assessment of the adversary's intentions. The plot shows the Allied troops' deficient capability to reject attack (20%), with a defender's Advantage Factor $v = -0.446$. Therefore, $P_{Germantroops} < 80\%$, matching with the real intention of the German Strategy on 27 April 1941.

This assessment has taken into account the following points:

- If the adversary needs a high probability of success to launch the attack, the adversary has a high risk of aversion (otherwise, a low risk of aversion). Since opponents are unaware of their enemy's aversion risk, this parameter is estimated.
- The risk of aversion is conditioned by the sizes of the armies and the uncertainty of the available information. In our case, the risk of aversion of German troops is low.

3.2.4. Scheduler Block

The evaluation of the alternative COA that could have been carried out to prevent the defeat of the Allies will be performed using decision trees, following the procedure depicted in Section 2.3.1, which considers the possible battle outcomes obtained from the predictive block.

Before the evaluation, the following should be taken into account: German troops occupied Crete island without numerical superiority, the effectiveness factor of the German troops was the cause for the Allied defeat as German paratroopers and mountain troops were better trained, motivated, and organized, as opposed to Allied troops, which were poorly equipped, worn out, poorly trained and organized by nationalities. From the above conclusions and taking into account the strategy defined in the previous Section 3.2.3, we will define a series of operational options that would avoid the defeat.

- Increasing lethality: Greek troops were poorly armed. This course is selected to increase the factor of lethality by supplying armaments.
- Defensive position, fortified terrain: The Allies were not prepared for the defense of the island, the maneuver of work, and the creation of fortified zones that would have prevented the island invasion.

This evaluation has taken into account the following points:

- In the case of a frontal attack, the ability to reject it is conditioned by the law 3:1 of terrestrial combat, i.e., the defender has an advantage factor of 3 to 1 whenever it is deployed on favorable terrain and with a defensive position. In other situations, the most appropriate ratio is 1.5:1, according to the research conducted by Davis [34]. Thus, applied to the evaluation, it means the increase of the allied force by a factor of 1.5.
- According to Strickland' [35], in most historical combats, the relations between initial and final concentrations of forces are relatively high, considering that the breakpoint of the battle (the end) takes place with attrition of forces >30% to a contender.

After the assessment, see Figure 11 where the different COAs are developed, Figure 12 where the scatter plot for the winning option is depicted, and Figure 13 where the scatter plot for the defeated option is depicted, the chosen COA is defensive position, fortified terrain concerning assuring conservative casualties and avoiding the occupation of Crete.

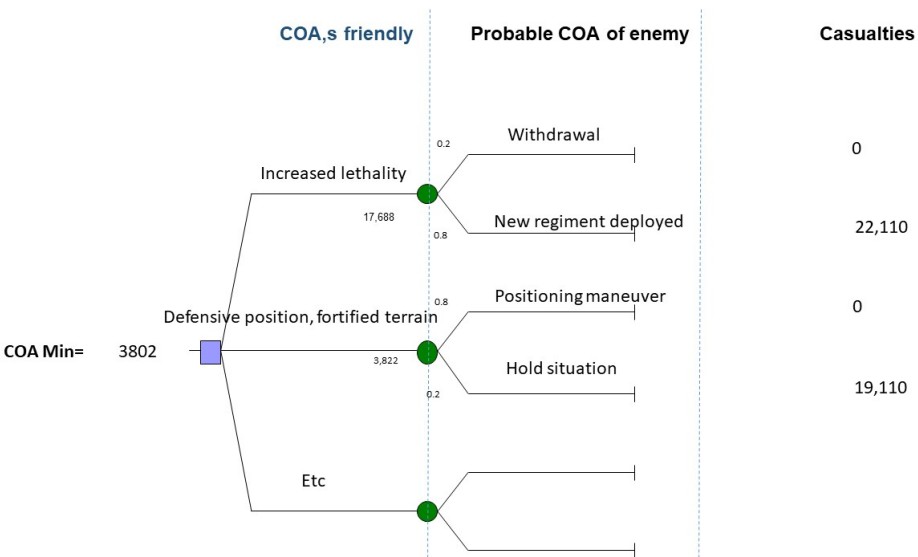

**Figure 11.** Development of the various courses of action, each Allie COA creates likely reaction alternatives for the German strategy quantified through its doctrine. The final expected value for each COA defines the best choice.

After this proof of concept, it was possible to experimentally test the evolution of events if other decisions had been made in the theater of operations, based on the ability to anticipate as an application in the automation of decisions high-level resolution.

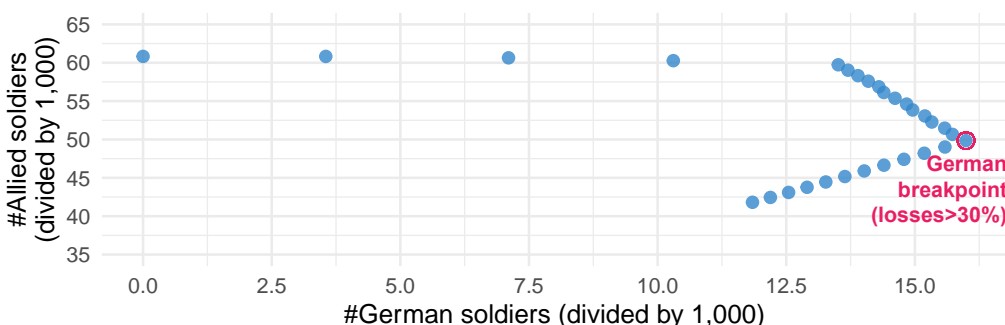

**Figure 12.** Scatter plot obtained from the predictive block for defensive position and fortified terrain as an alternative course by applying a 1.5:1 ratio for Allied troops. The selected COA prevents the invasion of the island (the German breakpoint on the 8th day defines the Allied victory).

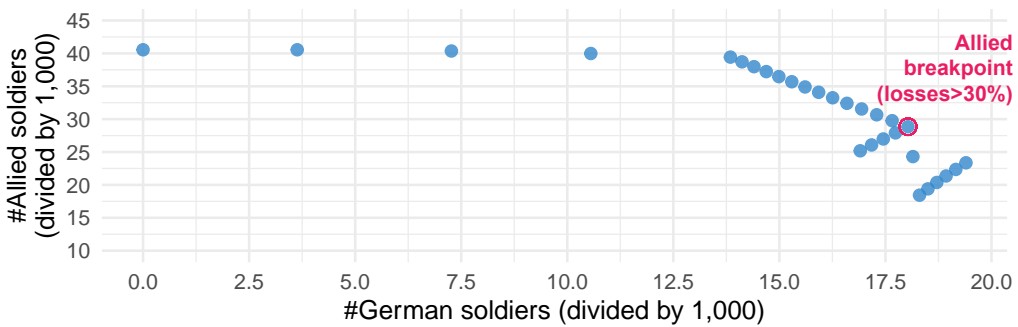

**Figure 13.** Scatter plot obtained from the predictive block for increasing effectiveness as an alternative COA, applying an increased effectiveness of 25%. This shows how the other COA does not prevent the invasion of the island (the Allied breakpoint on the 8th day defines the Allied defeat).

## 4. Conclusions and Future Work

Lanchester's classic work on battle dynamics modeling has inspired important research on the development of combat abstractions to support military decision-making under conditions of uncertainty, pursuing ways to achieve combat superiority. Nevertheless, it has been subject to the following criticisms: (a) it does not provide a fitting good enough for historical battle data, (b) it uses a constant lethality factor, (c) it deals with large battles with multiple types and phases as a whole, (d) it performs an oversimplistic one-sided treatment without taking into account opponent's capabilities, and (e) it cannot be used for disaggregated engagements.

To face those criticisms, this paper proposes a model focused on the types of decisions supported, how these types of decisions are made, and understanding the battle as a cause-effect process that evolves subject to changes and external actions. Thus, our framework removes the limitations of Lanchester's classic work by dynamically adjusting the factors that define the evolution of the battlefield, including learning mechanisms that optimize the capabilities of the architecture and, in short, the ability to improve decisions under uncertainty.

In the paper, we have provided empirical evidence showing that our framework fits battle trends adequately and can select the most appropriate COA. As a result, our approach contributes to existing research by supporting decision-making at a high command level.

Currently, our framework assumes that the cause-effect relationship of the battle is modeled. However, there may be a chaotic behavior in the final phases that makes such modeling difficult in some battles. We plan as future work to apply artificial intelligence techniques to overcome this problem. Additionally, we will consider incorporating into our model additional factors that may or not depend on the size of the forces. These factors could vary around fixed values as a function of the noise presented by confusion, momentum, and combat stress. Moreover, we will continue expanding the capabilities of some of the constituent blocks in the architecture. For example, in the case of the scheduler block, we are working on determining enemy disposition patterns that allow us to estimate detailed tactical intentions.

**Author Contributions:** Conceptualization: G.M.-C.; Data curation: G.M.-C.; Formal analysis: G.M.-C.; Funding acquisition: R.H. and C.C.; Investigation: G.M.-C.; Methodology: G.M.-C.; Project administration: R.H. and C.C.; Resources: R.H. and C.C.; Software: G.M.-C. and R.H.; Supervision: R.H.; Validation: G.M.-C. and R.H.; Visualization: G.M.-C. and R.H.; Writing—original draft: G.M.-C.; Writing—review and editing: G.M.-C. and R.H. All authors have read and agreed to the published version of the manuscript.

**Funding:** This work has been supported by the Community of Madrid under Grant S2018/NMT-4331 RoboCity2030-DIH-CM, and the Universidad Nacional de Educacion a Distancia under Project OPTIVAC Ref. 096-034091.

**Institutional Review Board Statement:** Not applicable.

**Informed Consent Statement:** Not applicable.

**Conflicts of Interest:** The authors declare no conflict of interest.

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
