# Peer review of "Automated Support for Battle Operational–Strategic Decision-Making"

_mathematics, doi:10.3390/math9131534_

Round 1

Reviewer 1 Report

It is, for me, an interesting and challenging subject.
I recognize scientific interest and the dynamic modelling proposal seemed to me
to be significantly innovative.
I wonder about its applicability to other combat scenarios,
more from the present days.

Author Response

Responding to the comments of Reviewer 1 by providing the attached pdf

Reviewer 2 Report

Dear Authors,

In fact, the text is not ready to be published in the present form. Some aspects should be improved to avoid misunderstanding, lack of clearness, and enhance the overall quality of the publication.

Please find my comments in the attached file.

Kind regards

Kind regards

Author Response

Responding to the comments of Reviewer 2 by providing the attached pdf

Reviewer 3 Report

The paper gives a very good introduction, and the methods used are OK.

The methods lack an explicit modeling of stochastics/noise, e.g., even a term in Eqs. (1) and (2) with their own coefficients.  There is no discussion of this in the paper?

Stochastics/noise is an important aspect of battle as well as of the use of regression methods.

Author Response

Responding to the comments of Reviewer 3 by providing the attached pdf

Round 2

Reviewer 2 Report

Dear Authors,

Thank you for resubmitting your manuscript. Now the structure of this paper and study results are presented in an appropriate style.

Kind regards